# Live Cell Monitoring of Phosphodiesterase Inhibition by Sulfonylurea Drugs

**DOI:** 10.3390/biom14080985

**Published:** 2024-08-10

**Authors:** Filip Berisha, Stefan Blankenberg, Viacheslav O. Nikolaev

**Affiliations:** 1Department of General and Interventional Cardiology, University Heart and Vascular Center Hamburg, University Medical Center Hamburg-Eppendorf, 20246 Hamburg, Germany; f.berisha@uke.de (F.B.);; 2Institute of Experimental Cardiovascular Research, University Medical Center Hamburg-Eppendorf, 20246 Hamburg, Germany; 3German Center for Cardiovascular Research (DZHK), Partner Site Hamburg/Kiel/Lübeck, 20246 Hamburg, Germany

**Keywords:** exchange protein directly activated by cAMP (Epac), glibenclamide, sulfonylurea (SU), tolbutamide, phosphodiesterase (PDE)

## Abstract

Sulfonylureas (SUs) are a class of antidiabetic drugs widely used in the management of diabetes mellitus type 2. They promote insulin secretion by inhibiting the ATP-sensitive potassium channel in pancreatic β-cells. Recently, the exchange protein directly activated by cAMP (Epac) was identified as a new class of target proteins of SUs that might contribute to their antidiabetic effect, through the activation of the Ras-like guanosine triphosphatase Rap1, which has been controversially discussed. We used human embryonic kidney (HEK) 293 cells expressing genetic constructs of various Förster resonance energy transfer (FRET)-based biosensors containing different versions of Epac1 and Epac2 isoforms, alone or fused to different phosphodiesterases (PDEs), to monitor SU-induced conformational changes in Epac or direct PDE inhibition in real time. We show that SUs can both induce conformational changes in the Epac2 protein but not in Epac1, and directly inhibit the PDE3 and PDE4 families, thereby increasing cAMP levels in the direct vicinity of these PDEs. Furthermore, we demonstrate that the binding site of SUs in Epac2 is distinct from that of cAMP and is located between the amino acids E443 and E460. Using biochemical assays, we could also show that tolbutamide can inhibit PDE activity through an allosteric mechanism. Therefore, the cAMP-elevating capacity due to allosteric PDE inhibition in addition to direct Epac activation may contribute to the therapeutic effects of SU drugs.

## 1. Introduction

Type 2 diabetes (T2D) is one of the most common diseases in industrial nations and is characterized by high blood sugar, insulin resistance and relative insulin deficiency [1,2,3,4]. It can lead to chronic complications and comorbidities, with its incidence continuing to rise in both Western and developing countries. During the development of insulin resistance and deficiency, the demand for insulin produced by the pancreas increases [5,6]. One therapeutic strategy to address this demand is to enhance insulin secretion from pancreatic β-cells [7]. This is achieved by the first class of oral antidiabetic drugs, the sulfonylureas (SUs), that are frequently prescribed to treat T2D [8,9]. They act by directly binding to the so-called SU-receptor (SUR), which is a regulatory subunit of the ATP-sensitive potassium (K_ATP_) channel. The binding of SUs to the SUR mediates the closure of this channel. Usually, high glucose concentration leads to membrane depolarization and Ca^2+^ influx, which trigger vesicle fusion and insulin release [10], whereas SUs increase insulin secretion by inhibiting the K_ATP_ channel [8]. In both diseased and healthy β-cells, 3′,5′-cyclic adenosine monophosphate (cAMP), as a second messenger, further increases insulin secretion, mainly by the activation of cAMP-dependent protein kinase (PKA) and exchange protein directly activated by cAMP (Epac) [10]. cAMP is a ubiquitous second messenger that regulates a high number of cellular processes, such as gene expression [11], cardiac contractility [12] and insulin secretion [13]. cAMP effects are limited by phosphodiesterases (PDEs), a class of enzymes that hydrolyze cAMP to AMP [14,15]. Epac is a guanine nucleotide exchange factor (GEF) for the Ras-like small guanosine triphosphatases Rap1 and Rap2 that is activated by the direct binding of cAMP [16,17,18]. Two major isoforms of Epac have been identified. Epac1 has just one cAMP binding site and is ubiquitously expressed, whereas Epac2 has an additional low-affinity cAMP-binding domain A and is predominantly expressed in the central nervous system, adrenal glands and pancreas [16,17]. Both are present in pancreatic islet cells [19] and Epac2 is involved in natural insulin secretion [20] by inhibiting K_ATP_ channels and activating calcium released channels, also called ryanodine receptors [21,22]. In addition, Epac2 can directly stimulate the docking and fusion of exocytotic vesicles [23,24].

Some reports have suggested that SUs can directly activate Epac2 [25] and that this effect is isoform-selective [26]. Experiments with Epac2 knockout mice supported the hypothesis that Epac2 plays a significant role in cAMP-dependent insulin secretion, indicating that the activation of Epac2 is required for maximum insulin secretion. However, Epac2 alone seems to be dispensable for the effect of SU drugs, as SUR1 knockout mice did not show increased insulin levels after SUs treatment [27]. Nenquin and Henquin used cAMP analogues and selective activators of Epac and PKA in pancreatic cells from SUR1 knockout mice, and both compounds were able to amplify insulin secretion [28]. However, in this study, SUs neither increased insulin secretion alone nor did they augment Epac2-induced amplification of insulin secretion in SUR1 knockout islets. The authors concluded that the direct activation of Epac2 does not seem to be involved in the action of therapeutic concentrations of SU drugs in β-cells [28].

Zhang et al. showed that SUs directly bind to and activate Epac2 as RapGEF. This finding was confirmed by Herbst et al. [26] and challenged by Tsalkova et al. [29]. Zhang et al. used COS cells expressing an Epac2-based fluorescent biosensor, which reports changes in Epac2 conformation based on Förster resonance energy transfer (FRET). Treatment with SUs led to a significant FRET response comparable to that of cAMP analogues. This was confirmed by Herbst et al., using a purified Epac2-based biosensor in vitro and in living cells. Furthermore, the use of cAMP analogues showed no competition with glibenclamide, suggesting that SUs bind to a binding site that is distinct from that for cAMP. On the other hand, Tsalkova et al. showed no activation of Rap1 by SUs in a classical RapGEF pull-down assay. In addition, isothermal titration calorimetry (ITC) could not reveal any interaction between SH and Epac2 [29]. However, Shibasaki et al. proposed that Epac2 contains a SU binding region located in its cyclic nucleotide binding domain A (CNBD-A), and that this region is not accessible to SUs if cAMP is absent due to the closed conformation of the second CNBD-B [30]. This would propose a synergistic activation of Epac2 by cAMP and SUs and could also explain the mentioned discrepancies between previous studies, making Epac an attractive therapeutic target for T2D [31].

Moreover, Takahashi et al. used molecular docking simulations to identify distinct binding sites for cAMP and SUs in Epac2 CNBD-A [32]. They confirmed these findings in FRET studies with an Epac2 CNBD-A-based FRET biosensor and using direct sulfonylurea-binding experiments. They concluded that SUs and cAMP cooperatively activate Epac2A through binding to CNBD-A and CNBD-B, respectively, and that the SU effect depends on the binding of cAMP. This contrasts with molecular docking simulations and mutant FRET sensor data by Herbst et al., which identified R447 adjacent to the CNBD-B as the crucial SU-binding amino acid. However, the idea of cooperativity between cAMP and SU binding provides a possible explanation for the discrepancies in the above-mentioned studies [32].

Another possible explanation for the observed discrepancies could be that SUs indirectly activate Epac by increasing the concentration of cAMP [33], since it has been shown that SUs can also inhibit PDE activity [34,35]. This presents yet another potential mechanism of SUs action, since PDE3B has been identified as one of the main regulatory enzymes involved in insulin secretion [36]. This assumption is supported by the observation that, in INS-1 cells, treatment with tolbutamide (TOL) and 3-isobutyl-1-methylxanthine (IBMX) each leads to a dosage-dependent PKA activation measured by FRET [37].

In this study, we used various available and newly developed FRET-based biosensors, containing either different fragments from Epac1 and Epac2 or PDE3 and PDE4 isoforms, fused to cAMP biosensors, to directly monitor PDE inhibition by SUs in living cells. We found that SU drugs can both inhibit PDEs and directly activate Epac-based biosensors. Therefore, both mechanisms may potentially contribute to their therapeutic effects.

## 2. Materials and Methods

### 2.1. Construction of Fluorescent Biosensors

To generate new biosensors for real-time monitoring of PDE inhibition by SU drugs, fragments containing sequences encoding for enhanced yellow fluorescent protein (YFP)-Epac2 from the Epac2-camps biosensor [38] and for enhanced cyan fluorescent protein (CFP) without the stop codon from the Epac1-camps-PDE4A1 biosensor [39] were amplified by PCR and cloned into the Epac1-camps-PDE4A1 vector between XbaI and BamHI using triple ligation to generate Epac2-camps-PDE4A1. PDE3B (a kind gift from Vincent Manganiello, Bethesda, MD, USA) was amplified by PCR and cloned into the Epac1-camps vector [38], between the NotI and XhoI restriction sites. CFP, containing a stop codon, was removed and replaced by CFP without a stop codon from ECFP-N1 (Clontech), between XbaI and XhoI, to generate Epac1-camps-PDE3B. This was repeated with Epac2-camps to generate Epac2-camps-PDE3B. To use different versions of Epac2 for monitoring the SU drug-induced conformational changes, the whole sequence, containing both cAMP binding domains, was amplified by PCR and cloned into the CFP-Epac1δDEP-YFP vector (a kind gift from Kees Jalink, Amsterdam, The Netherlands) to generate CFP-Epac2-YFP. Figure 1 shows an overview of the sensors that were generated and used in this study.

### 2.2. Cell Culture and In Vitro Measurements

The HEK293a cells (Invitrogen, Karlsruhe, Germany) expressing moderate amounts of endogenous β_2_-adrenergic receptor were maintained in Dulbecco’s modified Eagle’s medium containing 10% fetal calf serum, 100 U/mL of penicillin, 0.1 mg/mL of streptomycin at 37 °C, 7% CO_2_ and plated on 100 mm dishes. After 24 h, they were transiently transfected with 1 μg of a sensor DNA using the Effectene^®^ transfection reagent (Qiagen, Hilden, Germany). The cells were replated onto 24 mm coverslips 24 h after transfection. For in vitro measurements, cells were plated on 150 mm dishes and transfected using calcium phosphate precipitation. Cells were harvested 24 h after transfection in ice-cold 5 mM Tris, 2 mM EDTA, pH = 7.4 buffer, briefly ultraturraxed on ice, and centrifuged for 20 min at 278,900× *g*. The fluorescent spectra of the cytosol were measured using a spectrometer LS50B (PerkinElmer Life Sciences, Waltham, MA, USA) before and after adding various concentrations of SUs followed by 200 µM cAMP.

### 2.3. FRET Measurements in Living Cells

Coverslips were put into an Attoflour cell chamber (Invitrogen) 24 h after transfection, and FRET was measured in single cells using a microscope equipped with a Polychrome V monochromator (436 nm), ET436/30x excitation filter, DCLP455 dichroic mirror and a DV2 DualView (Photometrics), containing ET480/30 and ET535/40 emission filters and a dcxr505 beam splitter, as previously described [38]. Cells were kept in a physiological FRET buffer (containing NaCl 144 mM, 5.4 mM KCl, 2 mM CaCl_2_, 1 mM MgCl_2_, HEPES 10 mM and pH = 7.3) at room temperature and stimulated with the β-adrenergic receptor agonist isoproterenol, SUs and different PDE inhibitors.

### 2.4. Biochemical PDE Activity Assay

The HEK293 cells were transfected with the PDE4A1 expression plasmid, as described above. Cell lysates were prepared 24 h after transfection, by sonication and centrifugation in a buffer containing 40 mM Tris (pH = 8.0), 10 mM magnesium chloride and protease/phosphatase inhibitors. PDE activity assay was performed at room temperature in the same buffer with added 0.1% bovine serum albumin in the presence of various TOL concentrations and 10 µM of 2′-O-(N’-methylanthraniloyl)adenosine-3′,5′-cyclic monophosphate (MANT-cAMP), used as a fluorescent substrate as previously described [40]. Fluorescence was read at 450 nm peak upon 360 nm excitation for a total of 30 min using FlexStation 3 (Molecular Devices). PDE4 activity fraction was defined using 10 µM rolipram as a full selective inhibitor.

### 2.5. Statistical Analysis

The data were analyzed using Origin software and presented as means ± SE. Differences were tested using one-way ANOVA, followed by Sidak’s multiple comparison test.

## 3. Results

### 3.1. Measuring Direct Epac-Based Biosensor Activation in Living Cells

To study the effect of SUs on Epac2 containing both binding domains for cAMP (CNBD-A and CNBD-B) in addition to the already available biosensors, we generated a new construct containing full-length Epac2 fused between CFP and YFP. The whole Epac2 sequence (M1 to P993) was amplified by PCR and cloned into the CFP-Epac1δDEP-YFP vector instead of the Epac1 sequence (Figure 1). The resultant fusion protein CFP-Epac2-YFP was well expressed in transfected cells and showed a robust change in FRET signal in response to cAMP-elevating compounds. Next, we treated the cells expressing this biosensor with TOL, and the β-adrenergic receptor agonist isoproterenol was used as a positive control. Treatment with TOL resulted in a robust and rapid decrease in the FRET ratio, whereas this SU drug had no effect on CFP-(Epac1,δDEP)-YFP and Epac2-camps biosensors (Figure 2), suggesting the activation by SUs is isoform selective and the binding site is different from that of cAMP. Surprisingly, adding TOL also had a strong effect on the Epac1-camps biosensor but not on the longer version of Epac1 (CFP-(Epac1,δDEP)-YFP, Figure 2). Even at very high concentrations of TOL (2 mM), there was no FRET change detectable using Epac2-camps (Figure A1).

The absence of the TOL effect on Epac2-camps and the activation of Epac1-camps were not due to potential PDE inhibition by TOL and the resultant increase in cAMP levels, since various PDE inhibitors applied alone had no effect (Figure A2). Unexpectedly, the activatory effect of TOL was abolished in the Epac1-camps R279E mutant, which is insensitive to cAMP [42] (Figure A2), suggesting that either Epac1-camps produced a sensor-specific artifact upon TOL application or TOL might be able to bind to the R279 of Epac1-camps, generating a conformational change in the biosensor.

Interestingly, the Epac2A biosensor, which contains only a single cAMP binding domain, was not sensitive to TOL. In contrast, the Epac2-camps long and superlong biosensors, which are slightly extended versions of Epac2-camps containing only CNBD-B with additional 11 or 17 amino acids, could be transiently or more permanently activated by TOL, respectively, suggesting that the SU binding site might be located in a short amino acid region between E443 and E460 (Figure 3).

Based on data from Herbst et al. [26], we hypothesized that R447 could be a critical SU-binding residue and mutated it to alanine in the Epac2-camps superlong sensor. Indeed, this mutation could completely abolish the tolbutamide effect without affecting the ISO/cAMP-induced conformational change in the sensor molecule (Figure A3).

### 3.2. Direct Epac-Based Biosensor Activation In Vitro

The effects of TOL observed in living cells could be further confirmed in vitro using Epac1-camps and Epac2-camps biosensors in cell lysates. Glibenclamide (10 µM), a second-generation SU, or TOL, a first-generation SU drug, were added to the biosensors isolated from cell lysates. No significant change in FRET was observed after the addition of either SU to Epac2-camps. Subsequent treatment with cAMP was performed as a positive control to confirm the viability of the sensor (Figure 4). However, the Epac1-camps treatment with TOL led to a clear change in FRET (Figure 5).

### 3.3. Measuring Direct PDE Inhibition in Living Cells

Next, we expressed the already available sensors Epac1-camps-PDE4A1 and Epac1-camps-PDE3A in HEK293a cells to measure whether TOL can directly inhibit these PDEs. Since pancreatic cells mainly express PDE3B and Epac1-camps can be directly activated by SUs, which was unexpected and might be related to an artifact, additional biosensors based on Epac2-camps were developed to avoid mixed effects from the direct activation of Epac1-camps. Treatment with TOL led to clear concentration-dependent FRET responses on PDE3B in both Epac1-camps- and Epac2-camps-based sensors (Figure 6 and Figure 7).

TOL could also dose-dependently inhibit PDE4A1 activity, based on the measurements with a newly developed Epac2-camps-based biosensor (Figure 7). The FRET responses were somewhat higher in the sensors expressing Epac1-camps biosensors, where the direct activation of the Epac1-camps by TOL could potentially produce an additional effect, which could be ruled out using the Epac2-camps-based PDE fusion biosensors.

Our Epac-camps/PDE fusion biosensors can directly detect PDE inhibition in living cells based on a local increase in cAMP in the vicinity of the specific PDE used in the biosensor. To confirm that TOL indeed inhibits cellular PDE4A activity, we used a classical biochemical PDE assay that showed a similar concentration–response dependence to that obtained using live cell imaging, with half-maximal inhibition at a higher micromolar range (Figure 8A). To study the mode of enzyme inhibition, we performed this assay at various substrate concentrations, with and without TOL, to construct Lineweaver–Burk plots. In these plots, the linear fits of the enzyme kinetics, with and without TOL, intersect with the *x*-axis at a similar point, suggesting that TOL inhibits PDE4 by an allosteric mechanism (Figure 8B). In support of this notion, live cell imaging to measure PDE inhibition demonstrated the leftward shift in the concentration–response dependence to rolipram (Figure 8C), which is typical for allosteric PDE inhibitors [43].

## 4. Discussion

Sulfonylureas (SUs) act by inhibiting the ATP-sensitive K^+^ channel in pancreatic β-cells, which is being considered as the SU-receptor. In addition, further proteins have been identified as targets for SUs, which greatly add to their pharmacological effect. In recent years, Epac2 has been found to be another drug target for SUs. However, this has been the subject of controversial debate in the literature. Zhang et al. showed that SUs clearly had a reduced effect on insulin secretion in Epac2 knockout mice and concluded that Epac2 is needed for the maximum secretion of insulin [25]. Zhang et al. used a FRET-based biosensor that contained Epac2 that was activated after treatment with SUs similar to a cAMP analogue. This was confirmed by Herbst et al., using different biosensors [26] and challenged by Tsalkova et al., who showed that SUs were not able to activate Epac2 in a classical in vitro assay with purified Epac2 and Rap, which has been a well-established classical assay for the Epac Rap cascade [29]. Also, there was no interaction between SUs and Epac2 in ITC (isothermal titration calorimetry) or in equilibrium dialysis with radioactive SUs. On the other hand, the findings of Herbst et al. suggest the binding site of SUs in Epac2 to be vastly different from that of cAMP, involving some key residues in the hinge region of the CNBD-B such as R447. Furthermore, the observations made by Zhang et al. cannot fully exclude an indirect activation of Epac. It has been known for several decades that SUs can interact with different phosphodiesterases, although it is not clear how relevant this effect is under physiological conditions. It has been shown, as early as 1971, that tolbutamide can inhibit different PDEs [34,35] and, therefore, might potentially increase the concentration of intracellular cAMP. This would also have an impact on insulin secretion, since PDE3B is an important regulatory enzyme in pancreatic β-cells [36]. In INS-1 cells expressing the PKA activity biosensor AKAR3, it has previously been shown that both tolbutamide and PDE inhibitors, such as IBMX, can lead to a clear concentration-dependent FRET response [37].

In this study, we used various FRET-based biosensors to determine the specific impact of SUs on different PDEs and on conformational changes in the different domains of Epac1 and 2. To study the direct interaction between SUs and Epac we first used the biosensors Epac2-camps and Epac1-camps, which contain only one cAMP binding domain from the respective protein, in vitro. Neither tolbutamide nor glibenclamide lead to a FRET response in Epac2-camps, which meant no direct activation of this biosensor or that the binding domains of SUs and cAMP have very distinct locations in the Epac2 protein. Experiments in living cells showed similar findings. TOL showed no response in cells expressing Epac2-camps (Figure 2), even at very high concentrations (2 mM TOL, Figure A1). In contrast, cells expressing a biosensor containing the whole sequence for Epac2 (CFP-Epac2-YFP) showed a clear response to this SU, which supports the findings of Zhang and Herbst for SUs being able to directly activate Epac2. Since Epac2-camps, which has a higher sensitivity to cAMP, compared to full-length sensors [38], shows no reaction, an indirect activation through higher levels of cAMP seems very unlikely. Furthermore, the versions of the same biosensor (Epac2-long and Epac2-superlong) having slightly longer cAMP binding domains at the C-terminus, which start being responsive in terms of SU-induced activation, strongly support the hypothesis that the direct activation and the Epac2 binding domain B are connected to the area between the E443 and E460 of this protein (Figure 3). Indeed, in support of previous findings by Herbst et al., the mutation of R447 to alanine in our Epac2-superlong construct has abolished the TOL-induced conformational change in this biosensor (Figure A3). Surprisingly, Epac1-camps with only one binding domain from Epac1 showed a clear response to TOL. However, since a longer YFP-Epac1δDEP-CFP construct and Epac1-camps with a mutation (R279E), making it insensitive to cAMP, each showed no response to this SU (Figure 2 and Figure A3). These results suggest that Epac1-camps response to SUs might be due to some artifact typical for this particular sensor molecule or that the R279 residue, instead of R447, which is not present at this position in the Epac1 sequence, may respond with a conformational change upon SU binding to this particular biosensor, which does not take place in the longer versions of Epac1-based biosensors. On a critical and cautionary note, conformational changes measured by FRET, although often similar to those induced by cAMP, which is known to activate Epac, may not directly translate into an increase in Epac catalytic activity. This could be a potential explanation for the discrepancy of positive effects in FRET-based assays measured by many independent groups and the aforementioned lack of SU effect on Epac catalytic activity in the classical biochemical assays.

As early as in 1971, Brooker and Fichman [34], as well as Goldfine et al. [35], were able to show that SUs could inhibit PDE activity. At that time, they used classic biochemical in vitro assays and did not differentiate between the various PDE families, detecting, for example, ~40% reduction in PDE activity at 10 mM TOL. To study the hypothesis that there might also be an indirect Epac activation taking place via an increase in cAMP due to PDE inhibition, we used different, well-established biosensors designed to directly measure PDE inhibition in living cells [26]. We started by using our previously reported Epac1-camps-based biosensor for PDE4A inhibition, but then redesigned it based on Epac2-camps, since the latter cAMP biosensor itself is not affected by SUs. Based on this approach, we were able to demonstrate that SUs also have a clear inhibitory effect on the PDE4A and PDE3B subfamilies, directly measured as an increase in cAMP concentration in their vicinity. Tolbutamide (300 µM) showed a rapid response with a FRET-change having IC_50_ values of ~400 and ~700 µM for PDE4A and PDE3B, respectively. To compare these values with the actual TOL concentrations needed to inhibit the catalytic activity of PDE enzymes in vitro, we performed a classical PDE activity assay for PDE4A in the presence of different TOL concentrations, which showed a very similar IC_50_ value of ~300 µM (Figure 8A). Lineweaver–Burk plots and live cell imaging confirmed that TOL can inhibit PDE4 by an allosteric mechanism (Figure 8).

Herget et al. showed that rolipram has an IC_50_ value of 21 ± 1 nM for PDE4A1 and that cilostamide has an IC_50_ value of 0.37 ± 0.05 µM for PDE3A1 [39]. This indicates that for PDE inhibition, relatively high SU concentrations have to be used as compared to classical PDE inhibitors, with a measurable effect starting at ~100 µM TOL, which is in the therapeutically relevant range of this substance in patient plasma [35]. These numbers are much lower than in the studies from 1971, suggesting that SUs could have a much higher inhibitory effect on PDEs in living cells or in vivo than initially assumed.

## 5. Conclusions

In conclusion, SU drugs can both increase cAMP levels by inhibiting PDEs and directly induce conformational changes in Epac-based biosensors. While it is debatable if such conformational changes directly translate into an increase in Epac activity, our live cell imaging approach demonstrates that tolbutamide can directly inhibit PDE3 and PDE4 enzymes in high micromolar concentrations, which are within the therapeutically relevant range. Therefore, this mechanism may potentially contribute to the pharmacological effects of SU drugs in patients.

## Figures and Tables

**Figure 1 biomolecules-14-00985-f001:**
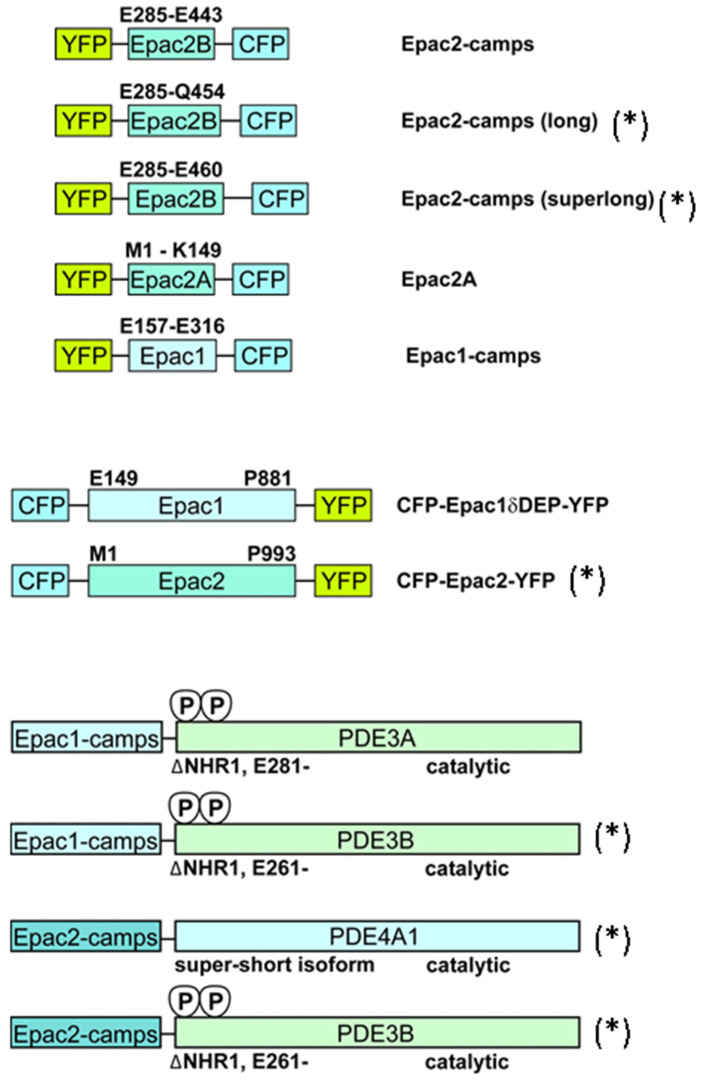
**Schematic overview of cAMP and cAMP/PDE biosensors**. Constructs marked with (*) were newly generated in this study. Epac1-camps and Epac2-camps developed previously [24] were used to exclude direct activation of single Epac cAMP-binding sites. CFP-Epac2-YFP includes the full-length Epac2 (M1 to P993), with both cyclic nucleotide binding domains (CNBDs) A and B. Epac2-camps long and superlong are slightly extended versions of Epac2-camps containing only the CNBD-B of Epac2 (encompassing amino acids E285-Q454 or E285-E460, respectively). Epac2A biosensor contains only a single CNBD-A from Epac2 (M1-K149), as previously described [41]. CFP-Epac1δDEP-YFP biosensor contains the N-terminally truncated Epac1 sequence, lacking only the first 148 amino acids encoding for the disheveled, Egl-10, pleckstrin (DEP) domain, which is responsible for membrane targeting. Fusions of Epac1-camps and Epac2-camps sensors to various PDE isoforms were developed as previously reported [31], with new sensors cloned for the Epac2-camps-based constructs fused to PDE3B and PDE4A1.

**Figure 2 biomolecules-14-00985-f002:**
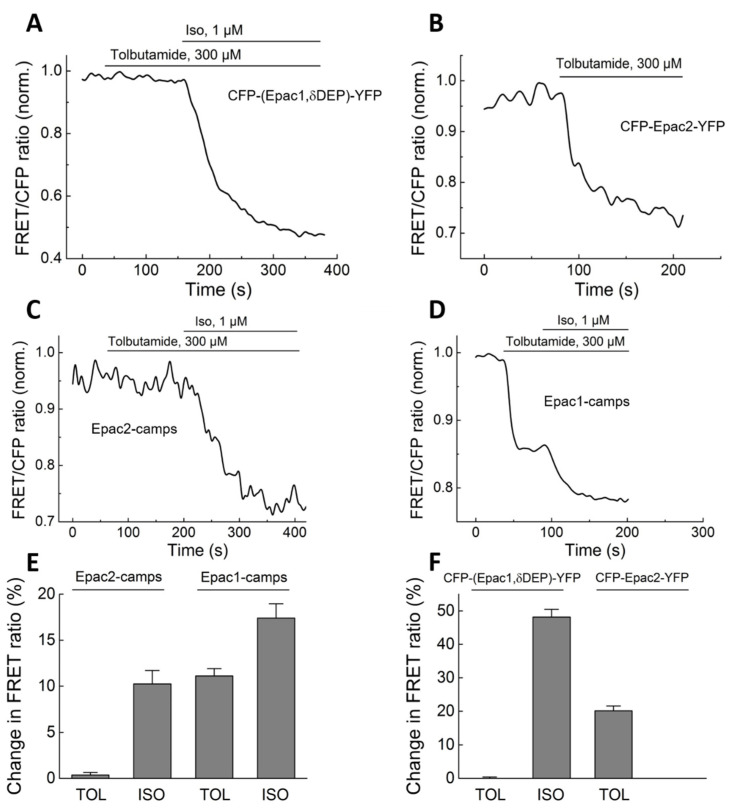
FRET measurements of Epac-based biosensor activation in living cells. HEK293a cells were transiently transfected with different cAMP sensors and treated with Tolbutamide (TOL) and Isoprenaline (ISO) as a control. CFP-(Epac1,δDEP)-YFP biosensor is not activated by TOL and responds only to increased cAMP after adding ISO as a positive control (**A**). Decrease in the FRET/CFP ratio reflects an increasing cAMP concentration or a similar conformational change in the biosensor construct. In cells expressing CFP-Epac2-YFP (**B**) but not Epac2-camps (**C**), treatment with TOL alone already leads to a change in FRET, suggesting that this SU drug induces a conformational change in the full-length Epac2-based CFP-Epac2-YFP biosensor. Surprisingly, Epac1-camps can also be rapidly activated by TOL (**D**). Representative curves (**A**–**D**) are from 4 to 6 independent experiments with multiple measured cells each (5–10). Data analysis (**E**,**F**) shows means ± SE from the following number of independent experiments, n = 5 for a: TOL 48.2 ± 2.3%; n = 5 for b: TOL 20.1 ± 1.5%; n = 6 for c: TOL 0.4 ± 0.3%; and n = 6 for d: TOL 11.1 ± 1.6%.

**Figure 3 biomolecules-14-00985-f003:**
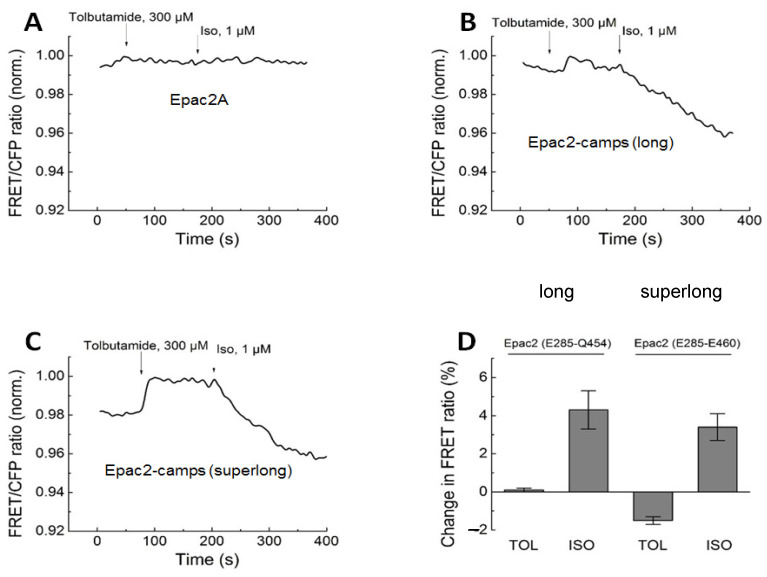
FRET measurements in cells expressing CNBD-A-based biosensor Epac2A and longer versions of the Epac2-camps biosensor. HEK293a cells were transiently transfected with Epac2A (**A**) or long and superlong versions of the Epac2 CNBD-B domain-based biosensor (**B**,**C**) shown in Figure 1 and treated with Tolbutamide (TOL) or Isoprenaline (ISO) as a control, as described in Figure 2. The long version of Epac2-camps responded to TOL with a small, clear, but transient response. The superlong version of the sensor, containing additional amino acids E443-E460, showed a positive change in the FRET signal, suggesting binding and TOL-induced activation. (**A**–**C**) Representative FRET traces and (**D**) data analysis of the FRET response to TOL and ISO, calculated after reaching a stable new baseline (n = 6–7).

**Figure 4 biomolecules-14-00985-f004:**
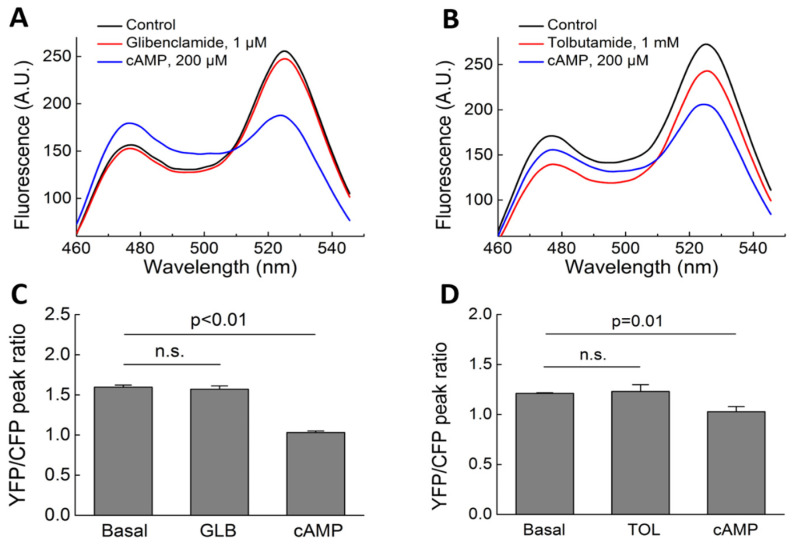
FRET measurements of Epac2-camps biosensor activation in vitro. Representative fluorescence emission spectra (n = 4) of the sensor recorded upon 436 nm excitation before and after adding Glibenclamide (**A**) or Tolbutamide (**B**). Decrease in the right acceptor peak fluorescence with a concomitant increase in the left donor peak fluorescence is indicative of a decrease in the FRET ratio, which reflects increasing cAMP concentration or a similar conformational change in the biosensor construct. There is a slight overall decrease in fluorescence intensity due to a slight dilution for both emission peaks, but no significant changes in FRET for both SU drugs (data analysis is in **C**,**D**). Mean values ± SE for measured YFP/CFP FRET ratios were for A: basal 1.60 ± 0.03 and after GLB 1.57 ± 0.04 (n.s., *p* = 0.35) and for B: basal 1.21 ± 0.01 and after TOL 1.27 ± 0.07 (n.s., *p* = 0.79). Subsequent stimulation with cAMP led to a dramatic decrease in FRET (blue curve) and served as a positive control. n.s., not significant by one-way ANOVA, followed by Sidak’s multiple comparison test.

**Figure 5 biomolecules-14-00985-f005:**
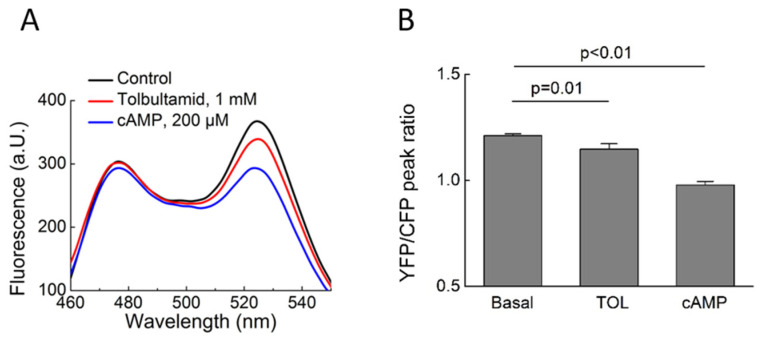
FRET measurements of Epac1-camps biosensor activation in vitro. Representative spectra (n = 3) were recorded before and after the addition of Tolbutamide (TOL, 1 mM) and show Epac1-camps biosensor activation by this drug. Representative spectra (**A**) and data analysis (**B**) showing mean values ± SE for measured YFP/CFP FRET ratios. *p* values are from one-way ANOVA, followed by Sidak’s multiple comparison test.

**Figure 6 biomolecules-14-00985-f006:**
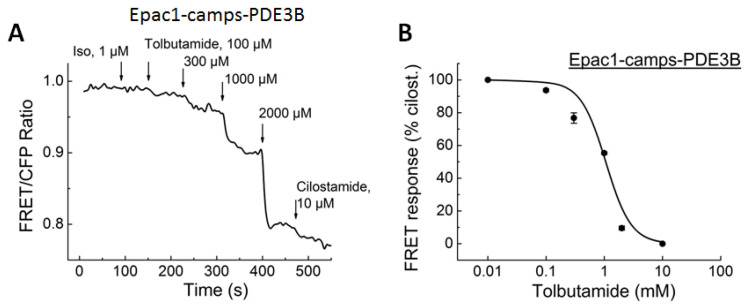
FRET measurements of TOL-induced PDE3B inhibition in cells expressing Epac1-camps/PDE3B fusion biosensor. HEK293a cells were transiently transfected with the biosensor and treated with increasing concentrations of TOL followed by the full inhibition of this PDE using 10 µM cilostamide. (**A**) Representative FRET tracing and (**B**) data analysis (n = 4). Decrease in the FRET ratio is indicative of direct PDE inhibition by tolbutamide, reported in real time by these biosensor constructs.

**Figure 7 biomolecules-14-00985-f007:**
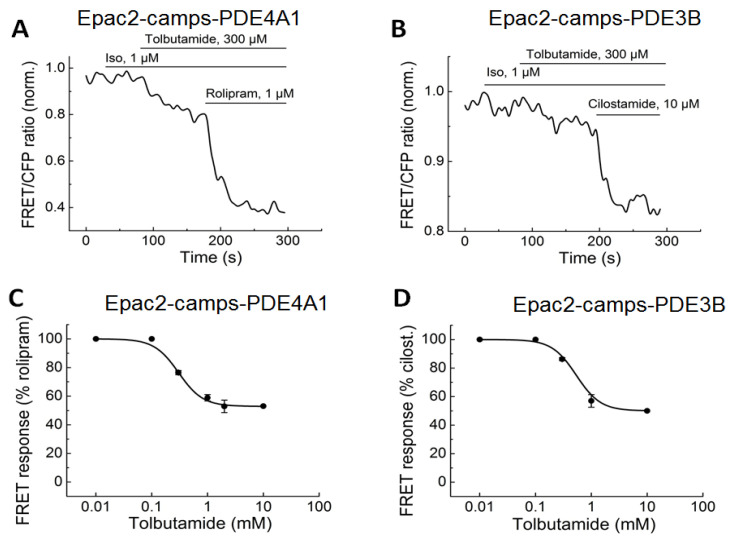
FRET measurements of TOL-induced PDE3B and PDE4A1 inhibition in cells expressing Epac2-camps/PDE fusion biosensors. HEK293a cells were transiently transfected with the respective biosensors and treated with increasing concentrations of TOL followed by the full inhibition of PDE3 using 10 µM cilostamide or PDE4 using 10 µM rolipram. (**A**,**B**) Representative FRET traces and (**C**,**D**) concentration–response dependence of the TOL effect on PDE3B and PDE4A (n = 6 independent experiments for c and n = 4 for d). Decrease in the FRET ratio is indicative of direct PDE inhibition by tolbutamide, reported in real time by these biosensor constructs.

**Figure 8 biomolecules-14-00985-f008:**
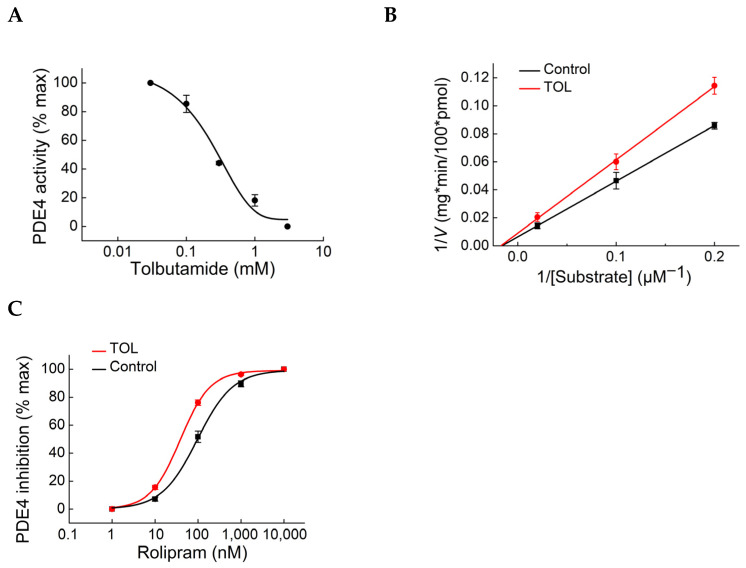
Measurement of PDE inhibition by tolbutamide in a classical PDE activity assay. (**A**) Cell lysates from HEK293 cells expressing PDE4A1 were treated with different concentrations of TOL, and PDE hydrolytic activity was assayed as described in the Materials and Methods Section using MANT-cAMP as substrate. The concentration–response dependence for TOL-induced inhibition of PDE4 activity is shown from several independent experiments (n = 5, values are means ± SE). (**B**) Lineweaver–Burk plots derived from measurements with different substrate concentrations performed using 50, 10 and 5 µM MANT-cAMP, with (red) or without (black) 300 µM TOL. Values are means ± SE (n = 4 each). (**C**) Concentration–response dependence for PDE4A inhibition by rolipram, measured using live cell imaging as described in Figure 7, in the absence (black) or presence (red) of 300 µM TOL. Values are means ± SE (n = 8 each).

## Data Availability

Data are contained within the article. Raw data and materials are available from the authors upon reasonable request.

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
