# Peer review of "Live Cell Monitoring of Phosphodiesterase Inhibition by Sulfonylurea Drugs"

_biomolecules, 2024, doi:10.3390/biom14080985_

Round 1

Reviewer 1 Report

Comments and Suggestions for Authors

General:

1) The general style and English syntax should be improved. Some ambiguous sentences are in need of substantial revision. A further source of reader frustration is the large number of undefined acronyms.  These should be carefully weeded out.

2) Spelling of the names of the drugs are the German version not English, and decimal comma is used in the text instead of the decimal point.

3) In the figure legends the various biosensor effects are presented without useful commentary as to what the difference between the observed effects of the sensors actually means. This is very difficult to follow for those who did not make or work with the constructs i.e. about 99.9% of prospective readers.  

Specific:

4) What are SUR(-/-) mice?   Line 55

5) Overall, the Introduction from line 55 to 78 belongs in the discussion.   

6) Line 85, it would facilitate the understanding of the paper by non-experts, if the purpose of the biosensor constructs listed in Fig1  would be briefly indicated. Yet another abbreviation not defined what is ∂DEP?

7) Regarding the inhibition of PDEs by sulfonylureas, the results are intriguing. However, it is straightforward to measure PDE activity and as the cDNAs are at hand these could be expressed and the respective activities analysed. Above all, it would be relevant to find out whether the effects are competitive substrate site inhibition or allosteric.  Provision of these would make the results with the EPAC probes much more incisive.

8) In the context of an inhibition of PDE activity by sulfonylureas, it is to some extent surprising that the short forms of Epac do not indicate an increase in cAMP levels. Is the indicator system “discerning” enough to show a potentiation of the cAMP response to isoproterenol by sulfonylureas as would be expected on the basis of functionally relevant PDE inhibitory efficacy?

Comments on the Quality of English Language

Requires improvement see above

Author Response

please see pdf document attached

Reviewer 2 Report

Comments and Suggestions for Authors

This report by Berisha et al applied genetic encoded FRET sensors containing fragments of Epac1 and Epac2 isoforms alone or fused to phosphodiesterases (PDEs) to monitor the effects of sulfonylurea (SU) drugs. The title of the study, “Mechanism of action of sulfonylurea drugs: direct Epac activation and phosphodiesterase inhibition”, is misleading. Changes in FRET signals suggest that SU induced measurable conformational changes in these sensors. To demonstrate direct activation or inhibition, the authors need to provide biochemical and cellular data showing that SU binds to EPAC2 and PDEs and consequently lead to the activation of EPAC2, i.e., increased guanine nucleotide exchange activity and inhibition of PDEs, i.e., decreased cAMP degradation. Moreover, relevant research related to Epac1 signaling and insulin secretion was not discussed and cited. For example, the authors did not discuss the rich literature that Epac2 plays an role in insulin secretion via regulating KATP channel activity (Kir6.2) by binding to SUR1 and subsequently leading to Ca2+-induced Ca2+ release (CICR), et al. More see below.

 Major issues:

 1.     All data presented in the study were solely based on ectopically expressed Epac-biosensors, which are proxies of conformational changes, and have no direct bearing on Epac or PDE functions.

2.     The nomenclatures of Epac in Figure 1 are not correct. Epac2A (isoform 1) is 1100 AAs long, Epac2B (isoform 2) is 993 AAs, Epac2C (isoform 3), not EPAC2A, is the one has only one cAMP binding domain. See PMID: 23861833.

3.     The authors missed several relevant findings. An elegant study by Nenquin and Henquin concluded that SU did not increase insulin secretion in sulphonylurea receptor-1 (SUR1) knockout islets, as would be expected if they were activating Epac2 directly, nor did SU augment the amplification of insulin secretion produced by Epac activator, suggesting SU is dispensable for amplification of insulin secretion by Epac2 activation and that direct activation of Epac2 is unimportant for the action of therapeutic concentrations of SU in β cells (PMID: 26584950).

4.     Moreover, a study by Takahashi et al (PMID: 24150255) showed SU and cAMP cooperatively activated Epac2A through binding to cNBD-A and cNBD. They also identified residues involved in SU binding. Surprisingly, this closely related work was not discussed and tested by the authors.

5.     Did the author try an Epac1-camps long version?

6.     The maximum change in FRET for CFP-Epac2-YFP in response to TOL is relatively small (~20%) when compared to that of CFP-(Epac1,δDEP)-YFP (~45%). The change in FRET for CFP-Epac2-YFP in response to ISO should also be included in Figure 2.

7.     The authors claim that TOL’s effects on Epac-sensors were due to PDE inhibition, again using EPAC-PDE fusion sensors. Why don’t the authors directly measure the intracellular cAMP to see if TOL treatment affects intracellular cAMP levels?

8.     If the effect of TOL on Epac1-camps was due to PDEs inhibition, why TOL had no effect on Epac2-camps?

9.     The maximum changes in FRET for Epac2-camps superlong shown in Figure 3 were very small (~2-4%). Also, the TOL and ISO led to opposite changes, unlike in the case of CFP-Epac2-YFP. What is the explanation? Don’t you expect to see similar changes (same direction) if TOL is inhibiting PDEs?

10.  The authors attribute the effect of TOL on Epac1-camps (>20% change in FRET) as an artifact. The fact that this effect was abolished by the R279E mutation argues otherwise.

 Minor points:

1.     Line #96: “fpr” should be “for”?

2.     Line #270: “und” should be “and”?

Comments on the Quality of English Language

Some of the sentences are difficult to understand. For example: "Experiments with knock out mice showed that Epac2 plays a significant role in cAMP-dependent insulin secretion, which indicates that activation of Epac2 is required for maximum insulin secretion, whereas the sole effect of Epac2 seems to have no effect, since in SUR1 (-/-)-mice no increase in insulin levels could be detected after SU-treatment."

Author Response

please see pdf document attached

Reviewer 3 Report

Comments and Suggestions for Authors

Line 33: “… rise not only in western but also in in developing countries.”

Line 38: “…directly binding to the so-called SU-receptor associated with ATP-sensitive a potassium.” SU-receptor (SUR) should be added.

Line 54: the following sentence is not quite clear: “…whereas the sole effect of Epac2 seems to have no effect, since in SUR1 (-/-)-mice no increase in insulin levels could be detected after SU-treatment [21].”

Line 79: “In this study, we used different available and newly developed FRET-based biosen-sors containing either different fragments from Epac1 and Epac2 or different isoforms of PDE3 and 4 fused to cAMP biosensors to directly monitor PDE inhibition by SU in living cells. We found that both cAMP elevating capacity due to PDE inhibition and direct Epac activation may contribute to the therapeutic effects of SU drugs.”

Line 155: No p values are not shown in figures 2 and 3.

Line 173: Authors stated that “SU binding site might be located in a short amino acid region between Q454 and E460 (Figure 3).” Having this data and the structure of TOL, it would have been great to see a bioinformatic docking assay. Alanine scanning assays, for example, could also be performed.

Line 186: Is not clear what is that the authors want to demonstrate with the Direct Epac-based biosensor activation in vitro. They only used 2 sensors. In addition, in Epac1-camps treatment with SU lead to a clear change in FRET (is this an artifact as well?)

Line 199: “in c and d). Mean values ± SE for measured YFP/CFP FRET ratios were for A: basal 1,60 ± 0,03 und”, und show be modified to and.

Line 314: authors have no evidence whatsoever about the statement: “both cAMP elevating capacity due to PDE inhibition and direct Epac activation may potentially contribute to the therapeutic effects of SU drugs.”

Being results in bibliography contradictory, the data presented in this manuscript do not add any clearness. Some interesting results should have been deeply analyzed, i.e. analyses of epac2 short an long versions, Epac1 artifact.

Comments on the Quality of English Language

some sentences are difficult to understand and there few typos to correct. 

Author Response

please see pdf document attached

Round 2

Reviewer 1 Report

Comments and Suggestions for Authors

The authors have addressed all of my comments in an adequate manner, except for their contention that :"This assay is based on using labeled cAMP as a substrate for which the inhibitor is competing, decreasing hydrolytic activity of this PDE. So we believe it is a competitive inhibition."  This is not a matter of belief. A Lineweaver-Burk reciprocal plot with three or four concentrations of inhibitor should resolve the issue.  If sulfonylureas are allosteric inhibitors, this would be an important finding and further enhance the iimpact of this study. 

Author Response

We thank the Reviewer for this helpful suggestion. To study the mode of enzyme inhibition, as suggested by the Reviewer, we have not performed PDE activity assays at various substrate concentrations with and without TOL to construct Lineweaver-Burk plots (please see the new Figure 8b). In these plots, linear fits of the enzyme kinetics with and without TOL intersect with x-axis at a similar point, suggesting that TOL indeed inhibits PDE4 by an allosteric mechanism. In support of this notion, live cell imaging to measure PDE inhibition demonstrated the leftward shift of the concentration-response dependency to rolipram (see new Figure 8c) which is typical for allosteric PDE inhibitors [43].

Reviewer 3 Report

Comments and Suggestions for Authors

Manuscript has been greatly improved.

Parragraph on line 254 is still confusing,

Based on data from Herbst et al. [26] we hypothesized that R447 could be s critical SU binding residue and mutated it to alanine in the Epac2-camps superlong sensorR. Indeed, this mutation could completely abolish tolbutamide effect without affected ISO/cAMP induced conformational change in the sensor molecule (Figure A3). 

Comments on the Quality of English Language

minor issues around the manuscript

Author Response

We thank the reviewer and apologise for this confussing phrase. We have now corrected two typos in this sentense to make it better readable "Based on data from Herbst et al. [26] we hypothesized that R447 could be a critical SU binding residue and mutated it to alanine in the Epac2-camps superlong sensor. Indeed, this mutation could completely abolish tolbutamide effect without affecting ISO/cAMP induced conformational change in the sensor molecule (Figure A3)."